# PCSK9 Inhibitors in Cancer Patients Treated with Immune-Checkpoint Inhibitors to Reduce Cardiovascular Events: New Frontiers in Cardioncology

**DOI:** 10.3390/cancers15051397

**Published:** 2023-02-22

**Authors:** Vincenzo Quagliariello, Irma Bisceglia, Massimiliano Berretta, Martina Iovine, Maria Laura Canale, Carlo Maurea, Vienna Giordano, Andrea Paccone, Alessandro Inno, Nicola Maurea

**Affiliations:** 1Division of Cardiology, Istituto Nazionale Tumori-IRCCS-Fondazione G. Pascale, 80131 Napoli, Italy; 2Servizi Cardiologici Integrati, Dipartimento Cardio-Toraco-Vascolare, Azienda Ospedaliera San Camillo Forlanini, 00152 Roma, Italy; 3Department of Clinical and Experimental Medicine, University of Messina, 98122 Messina, Italy; 4U.O.C. Cardiology Department, Ospedale Versilia, Lido di Camaiore (LU), 55049 Camaiore, Italy; 5Department of Neurology, University of Salerno, 84084 Fisciano, Italy; 6Medical Oncology Department, IRCCS Ospedale Sacro Cuore Don Calabria, 37024 Negrar di Valpolicella, Italy

**Keywords:** cardio-oncology, cardioprotection, cancer, immune-checkpoint inhibitors, atherosclerosis, cholesterol, inflammation

## Abstract

**Simple Summary:**

Atherosclerosis is a critical cardiovascular disease associated with the use of immune checkpoint inhibitors (ICIs). Proprotein convertase subtilisin/kexin type 9 (PCSK9) is a key orchestrator of atherosclerotic process and it is also involved in cancer progression and immune-resistance. In this context, data from recent meta-analysis and cardiovascular outcome trials associate PCSK9 levels to reduced ICIs-related cancer responsiveness and to high risk of atherosclerotic cardiovascular diseases. This review summarizes the pleiotropic effects of PCSK9 in heart failure, atherosclerosis, and cancer immune recognition, and outlines its ability to represent a new pharmacological target in patients who develop ICIs-related atherosclerosis to reduce cardiovascular mortality and to improve overall survival.

**Abstract:**

Cancer patients treated with immune checkpoint inhibitors (ICIs) are exposed to a high risk of atherosclerosis and cardiometabolic diseases due to systemic inflammatory conditions and immune-related atheroma destabilization. Proprotein convertase subtilisin/kexin type 9 (PCSK9) is a key protein involved in metabolism of low-density lipoprotein (LDL) cholesterol. PCSK9 blocking agents are clinically available and involve monoclonal antibodies, and SiRNA reduces LDL levels in high-risk patients and atherosclerotic cardiovascular disease events in multiple patient cohorts. Moreover, PCSK9 induces peripheral immune tolerance (inhibition of cancer cell- immune recognition), reduces cardiac mitochondrial metabolism, and enhances cancer cell survival. The present review summarizes the potential benefits of PCSK9 inhibition through selective blocking antibodies and siRNA in patients with cancer, especially in those treated with ICIs therapies, in order to reduce atherosclerotic cardiovascular events and potentially improve ICIs-related anticancer functions.

## 1. Introduction

Immune checkpoint inhibitors (ICIs) are increasingly used in oncology to treat multiple malignancies, including melanoma, non-small cell lung cancer, metastatic breast cancer, and others [1,2]. In most cases, ICIs are monoclonal antibody antagonists of programmed death-ligand 1 (PDL-1) or programmed cell death protein 1 (PD-1) or cytotoxic T-Lymphocyte Antigen 4 (CTLA-4) which are the main drivers of peripheral immune tolerance, even towards tumors [3,4]. Briefly, ICIs antagonize the inhibition of lymphocyte uptake against tumors, resulting in a lymphocyte-mediated anticancer effect [4]. Recent trials associate ICIs with radiotherapy [5], standard chemotherapy (anthracyclines or platinum-based anticancer drugs) [6,7], targeted therapies (HER-2 blocking agents, TKi and others) [8], and combination therapies (i.e., PD-1 and CTLA-4 blocking agents) [9,10]. In brief, combination therapies involving ICIs and standard chemotherapies increase lymphocytic infiltration in neoplastic tissue in a pro-inflammatory microenvironment, which make CD56+ and CD3− large granular lymphocytes more reactive against tumor cells [11,12]. However, ICIs therapies are associated with a broad spectrum of endocrine diseases and, albeit, relatively rare cardiovascular side effects [13,14], including myocarditis [15], vasculitis [16], inflammatory endocrinopathies [17], mucositis [18], and arthritis [19]. The main mechanisms of cardiotoxicity are not deeply understood, but NLRP-3/IL-1overexpression, My-D88/TLR4, and cytokine-mediated pathways are still considered to be key orchestrators [20,21] in ICIs-mediated side effects in preclinical and clinical models [21]. Very recently, atherosclerosis has emerged as a new considerable ICI-s mediated side effect in cancer patients [22,23]. Briefly, exposure to PD-1 or PDL-1 or CTLA-4 blocking agents increases VCAM-1 and ICAM-1 expression in luminal membrane of vascular endotheliocytes, thus stimulating IL-1, IL-6, and TNF-α levels associated with instability of the atherosclerotic plaque. Interleukins 1 and 6 and TNF-α induce LDL uptake and their oxidation to OX-LDL in endothelial cells [24,25,26] (Figure 1).

A new key driver of the atherosclerosis process is proprotein convertase subtilisin/kexin type 9 (PCSK9) [27]. Briefly, PCSK9 is a protein with key roles in hepatic low density lipoprotein (LDL) homeostasis, reducing the LDL-receptor density in hepatocytes [28]. When PCSK9 binds LDL receptors, it prevents the correct recycling on the cell membrane after the natural bond with LDL particles. This effect increases plasma LDL levels and its associated cardiovascular risk [29]. PCSK9 inhibitors (antibody-based blocking agents and miRNAs) are currently used in clinical practice to reduce LDL levels in high risk cardiovascular patients, intolerant to statins, in order to reduce cardiac risk [30]. However, extrahepatic functions of PCSK9 are increasingly studied in the last three years, involving cardiomyocyte [31], endotheliocyte [32], macrophage, and cancer cell metabolism [33], shedding light on possible therapeutic uses of PCSK9 inhibitors to new off-label clinical applications in cardio-oncology. This review assesses the current knowledge about how PCSK9 interacts with the immune environment in cancer tissue and how PCSK9 inhibitors could be beneficial in patients treated with ICIs in primary prevention of atherosclerosis. First, we describe how PCSK9 is involved in cancer progression and immune escape. Then, we review current knowledge on how PCSK9 inhibitors reduce atherosclerosis initiation and progression in patients without cancer and the main molecular pathways involved.

## 2. ICIs Therapy, PCSK9, and Risk of Atherosclerotic Cardiovascular Diseases

### 2.1. ICIs-Mediated Atherosclerosis

The use of ICIs can change the peripheral immune tolerance in different tissues, exposing patients to neuro-inflammatory diseases, visceral obesity, atherosclerosis, and leptin resistance [34]. Preclinical models with deactivating mutations of PD-1 or PDL-1 or CTLA-4 genes are more exposed to atherosclerotic plaques characterized by high levels of VCAL-1, ICAM-1, galectine-3, oxLDL, high macrophage density, and pro-inflammatory cytokines [35,36]. In brief, the lack of these immune repressor proteins increases atherogenic phenotype mediated by a high uptake of CD3+/CD4+ lymphocytes and CD3+/CD8+ lymphocytes in atherosclerotic plaque [37]. These findings led to the hypothesis that ICIs could increase the risk of atherosclerotic cardiovascular diseases (ASCD) in cancer patients [38] (Figure 1). Recent clinical studies have confirmed this hypothesis: patients treated with ICIs for two years have a three times higher risk of developing atherosclerosis compared to other therapies [39]. There are additional observational studies associated with the high risk of unstable atherogenic plaques in cancer patients treated with ICIs compared to the general population [40]. However, more in-depth studies should be performed to clarify if combinatorial anti-PD-1 and CTLA-4 therapies could affect the ASCD risk more than a monotherapy regimen.

### 2.2. PCSK9 Role in Atherosclerotic Pathogenesis

A close correlation between atherosclerosis and PCSK9 has also been observed [41]. In a recent meta-analysis [42], eleven studies in patients with CVD were analyzed. Notably, patients with established CVD and high PCSK9 levels had a 52% higher risk of future total cardiovascular events than those with low PCSK9 concentrations. Patients with high levels of PCSK9 experienced more than 20% of cardiovascular events compared to low levels [43]. Mechanistically, PCSK9 plays a key role in platelet aggregation and adherence to endothelial cells, endothelial dysfunctions, and atherosclerosis [44]. Genetic studies associated loss of function mutation in PCSK9 gene with high LDL levels and high rates of heart failure [45]. Other studies associated high serum PCSK9 levels to a more necrotic core of coronary atherosclerosis independently from LDL levels [46]. These data indicate extra hepatic roles of PCSK9 that do not involve cholesterol homeostasis. However, Vlachopoulos et al. do not associate PCSK9 levels to cardiovascular events in high-risk patients; they only do so in the general population [47]. Additionally, other research groups associated high PCSK9 levels to patients with stable CVD, but not in patients with ACS [48]. However, there are several other associations between PCSK9 and cardiovascular diseases. First, during myocardial infarction, PCSK9 serum levels are upregulated due to pro-inflammatory processes [49]. Moreover, other studies demonstrated that circulating PCSK9 levels rapidly rise after the initiation of statin therapy [50], and there is a sustained increase throughout statin use. Furthermore, it has recently been observed that PCSK9 could reduce the efficacy of statin treatment in patients with high cardiovascular risk, by reducing the LDL receptor density on hepatocytes [51]. Furthermore, a recent gender study concluded that white people and Asian people have variants in the PCSK9 gene that may or may not be associated with different LDL concentrations [52]. The main cardiovascular outcomes trials, which will be discussed in the next paragraph, called Evaluation of Cardiovascular Outcomes After an ACS During Treatment With Alirocumab (ODYSSEY) [53] and Further CV Outcomes Research with PCSK9 Inhibition In subjects With Elevated Risk (FOURIER) [54], analyzed cardiovascular benefits of PCSK9 inhibitors in non-cancer patients, and recruited mostly only white patients. Being a minority in the makeup made up of ascitic patients, it is probable that the effects of PCSK9 inhibitors in white people are not the same due to variations in the genetic profile of different races [54].

### 2.3. PCSK9 in Cardiomyocyte and Endothelial Metabolism

The PCSK9 gene is located on chromosome 1p32.3 that is expressed in several organs, including the liver, kidneys, small intestine, heart, and cancer cells [55]. The encoded protein is of 692 aminoacids, characterized by three domains: signal domain, pro-domain, and V domain (Figure 2). Interestingly, the key process for maturation and secretion of active PCSK9 is secondary to the cleavage of pro-domain in the S 38 and P47 region [56]. The first clinical evidence on the association between PCSK9 and CV events are based on loss of function or gain of function in PCSK9 gene: patients with gain of function mutations of PCSK9 gene experienced high levels of serum LDL-C, a reduction in LDL receptor levels on hepatocytes (by more than 35%), and premature cardiovascular events compared to non-mutated patients [57]. Contrary, loss of function mutations are associated with low serum levels of PCSK9 and more than 40% reduction of LDL-C with consequent risk reduction in incidences of ischemic heart diseases [57]. PCSK9 is also expressed in cardiomyocytes [58]. Very recent studies described PCSK9 functions in cardiomyocyte autophagy, pyroptosis, ferroptosis, and apoptosis [58]. Detrimental events such as hypoxia, acute inflammation, and heart failure can induce different types of cardiomyocyte cell death through PCSK9 overexpression [59]. Specifically, high cholesterol and insulin levels induce overexpression of PCSK9, which activates DRP-1 in mitochondria triggering autophagy [60]. Moreover, PCSK9 exacerbates mitochondrial ROS production, resulting in the activation of the LKB1-AMPK pathway [61]. Moreover, in cases of OX-LDL intake in the cardiomyocyte, PCSK9 overexpression causes mitochondrial DNA damage, resulting in the activation of NLRP3 inflammasome/Caspase-1/Interleukin-1 pathway [62]. This process activates cardiomyocyte pyroptosis. Increased serum levels of IL-1 and IL-6 directly result in PCSK9 overexpression, which also induces cardiomyocyte apoptosis via caspase-9 and 3 [63]. Another biochemical mechanism of PCSK9 cardiac and endothelial toxicity is mediated by lipid peroxides [64]. Lipid peroxidation, which can be induced by acute inflammation, smoking, and some chemotherapeutic drugs such as anthracyclines, leads to the formation of MDA and 4-HNA by the Fenton reaction [65,66]. PCSK9 activates the Fenton reaction, whose gene expression is activated by the over-intake of Fe3+ in cardiomyocytes and endotheliocytes [66]. Furthermore, intracellular overexpression of PCSK9 results in decreased FFA uptake and utilization through a specific mechanism: PCSK9 competes with FFAs for binding to the CD36/FAT receptor and reduces their membrane recycling, resulting in an increase in systemic FFA levels and a significant reduction in fatty acid beta oxidation and the Krebs cycle [67]; this is the primary driver of atherosclerosis and cardiomyocyte injury processes. Moreover, it is conceivable that PCSK9 has a role in heart failure, hypertrophy, and cardiac fibrosis. These effects could be mediated by TNF-a and some chemokines; in fact, TNF-a upregulates PCSK9 gene expression via Peroxisome proliferator-activated receptors (PPARs) and PPAR gamma coactivator-1a (PGC1) pathways [68].

## 3. PCSK9i in Cardiovascular Outcome Trials

Current guidelines of the American College of Cardiology and American Heart Association recommend the addition of non-statin cholesterol-lowering therapies for patients at very high risk of major adverse cardiovascular events (MACE) when LDL-C levels remain ≥ 70 mg/dL [69]. About half of coronary heart disease patients on moderate- or high-dose statin therapies reduce LDL-C less than 70 mg/dL [70]. In this category of patients, there could be an additional clinical benefit deriving from the addition or substitution with other cholesterol-lowering agents, such as PCSK9 inhibitors, and this is a point of discussion in cardiology. Some randomized clinical trials have demonstrated how the administration of PCSK9 inhibitors, evolocumab, alirocumab, bococizumab, or inclisirian, as monotherapy or in combination with statins can reduce systemic levels of atherogenic and pro-inflammatory lipoproteins [71,72]. Notably, PCSK9i significantly reduces LDL-C and up to 25% lipoprotein a (Lpa) [73]. Effects on LPa levels are of particular interest in cardiology, considering its well established strong atherogenic, pro-inflammatory, and pro-thrombotic effects [74]. A recent trial concluded that among patients with advanced stable coronary artery disease, Lp(a) is associated with accelerated progression of coronary low-attenuation plaque (necrotic core) [75]. This may explain the association between Lp(a) and the high residual risk of myocardial infarction, providing support for Lp(a) as a treatment target in atherosclerosis [76]. The most known CVOT including PCSK9i are the FOURIER and ODYSSEY Outcomes trials. Patients with established cardiovascular disease or acute coronary syndrome (ACS) and elevated levels of LDL-C, non–high-density lipoprotein cholesterol, or apolipoprotein B were enrolled in these studies [77]. Patients treated with PCSK9i reduced MACE, coronary heart disease, peripheral artery disease, and venous thromboembolic events. Some of these beneficial effects were also associated with LPa reductions [78]. A brief description of the clinical benefits of each PCSK9i is provided below and summarized in Table 1:-*Evolocumab*

Patients treated with evolocumab halved LDL cholesterol values [79]. Furthermore, a recent meta-analysis showed that patients treated with evolocumab also had a slight but significant increase in HDL cholesterol; on the other hand, total cholesterol levels decreased by about 35–37% but with great heterogeneity of age and sex [80]. Evolocumab was not effective in reducing triglyceride levels. Regarding adverse cardiac events, a recent meta-analysis showed that evolocumab reduced myocardial infarction by 27% and stroke by approximately 21% [81]. A further interesting finding concerns unstable angina requiring revascularization which was reduced by 16% in patients treated with evolocumab vs. placebo. A further recent meta-analysis showed that evolocumab treatment reduces composite or CV death, myocardial infarction, stroke, and unstable angina by 15%. Moreover, in HUYGENS (High-Resolution Assessment of Coronary Plaques in a Global Evolocumab Randomized Study) study [82], evolocumab administration stabilized coronary plaque, resulting in regression of atheroma volume in patients with ACS.
-*Alirocumab*

Alirocumab is an additional PCSK9 inhibitor that has been extensively studied in cardiovascular outcome trials [83]. A recent meta-analysis showed that patients treated with alorocumab increased HDL cholesterol levels by approximately 5% [84]. Similarly to evolocumab, alirocumab reduces total cholesterol levels without affecting triglyceride levels. Furthermore, alirocumab, similarly to evolocumab, reduced composite of MI, stroke, unstable angina, and CV death. In a biweekly administered regimen, Alirocumab lower LDL-C by 45 to 60% depending on the applied dose (75 mg vs. 150 mg) and by ~ 50% while given monthly in a 300-mg dose [85]. Moreover, we now have evidence that evolocumab and alirocumab treatment, on top of statins, in patients with ACS modifies coronary plaque properties, leading not only to a significant thickening of the fibrous cap, thereby stabilizing it, but also resulting in regression of atheroma volume in PACMAN-AMI (Effects of the PCSK9 Antibody Alirocumab on Coronary Atherosclerosis in Patients With Acute Myocardial Infarction) study [86].
-*Bococizumab*

Bococizumab is an additional PCSK9 blocking monoclonal antibody which, unlike the other antibodies, has shown beneficial effects on MACE only in high-risk patients [87]. Two large studies called SPIRE I [88] and II [89] enrolled more than 27,000 low-risk and high-risk patients (LDL 70 and 100 mg/dl, respectively) treated with bococizumab (150mg) twice a week. In SPIRE I [88], bococizumab showed no significant effects on composite of myocardial infarction, stroke, hospitalization for unstable angina requiring urgent revascularization, and cardiovascular death. IN SPIRE II, [89] however, bococizumab reduced the primary endpoint of the SPIRE studies by 11%. However, a serious problem occurred in these patients, namely the production of antibodies to bococizumab, which resulted in very frequent injection site adverse reactions [90].
-*Inclisiran*

Inclisiran is a novel, small, interfering RNA aimed to target PCSK9 [91]. In detail, the molecule is a modified double-stranded RNA conjugated to triantennary N-acetylgalactosamine (GalNAc). Liver cells are rich in the asialoglycoprotein receptor that binds to GalNAc, therefore, after subcutaneous administration, inclisiran accumulates significantly in the liver [92]. Inclisiran lowered PCSK9 and LDL-C levels in a dose-dependent manner [93]. After 2 doses of inclisiran, LDL-C was reduced by up to 53% at day 180 [94]. Three large, randomized trials evaluated the cardiovascular efficacy of incisiran; these were called ORION 9–10 and 11 trials [95,96]. In the ORION-9 trial, HeFH patients halved LDL cholesterol after approximately 500 days of treatment with inclisirian. In ORION 10 and 11, LDL cholesterol levels were reduced by approximately 52 and 49% over 500 days of treatment with inclisirian. Furthermore, in all trials, inclisirian also significantly reduced levels of total cholesterol, apolipoprotein B, triglycerides, and lipoprotein (a), thereby reducing many cardiovascular risk factors [97]. Another cardiovascular outcome trial, currently underway, called ORION-4 (NCT03705234) [98], recruited 15,000 patients with ASCVD treated with inclisirian (300 mg) administered as a SC injection at randomization, (3 months and then every 6 months) or placebo aimed to evaluate any reductions in coronary heart disease (CHD) death; these were myocardial infarction, fatal or non-fatal ischemic stroke, or urgent coronary revascularization procedures [98].
cancers-15-01397-t001_Table 1Table 1Main cardiovascular benefits derived from PCSK9i therapies in randomized controlled trials.DrugChemistryCardiovascular BenefitsReferencesEvolocumabAntibody-50% reduction in LDL levels[79]

-35–37% reduction in total cholesterol; increases in HDL levels-27% reduction of myocardial infarction; 21% reduction of stroke-15% reduction in composite or CV death, myocardial infarction, stroke, and unstable angina events-Stabilization of coronary plaque in patients with ACS[80][81][81][82][83]AlirocumabAntibody-5% increase in HDL levels-45/60% reduction in LDL levels-Reduction in composite of MI, stroke, unstable angina, and CV death-Stabilization of atherosclerotic plaque and regression of atheroma volume in patients with MI[84][85][85][86]BococizumabAntibody-No significant effects on composite of MI, stroke, hospitalization for unstable angina requiring urgent revascularization, and cardiovascular death[88]

-11% reduction in composite of MI, stroke, hospitalization for unstable angina requiring urgent revascularization, and cardiovascular death[89]InclisiransiRNA-53% reduction in LDL levels[94]

-52 and 49% reduction in LDL levels-Significant reduction of total cholesterol, Apolipoprotein-B, Triglycerides, and Lipoprotein (a)[95,96][97]


## 4. PCSK9i in Oncology: Mechanisms and Potential Application

Inhibitors of PCSK9 have been approved for the treatment of atherosclerotic cardiovascular diseases associated with hypercholesterolemia, however, a central role on immune tolerance in oncology has recently been investigated [99]. Mechanistically, PCSK9 inhibits the recycling of major histocompatibility complex type I (MHCI) to the cell surface by promoting MHC I degradation (Figure 3) [100]. The inhibition of PCSK9 increases the expression of MHC I on the tumor cell surface, promoting intratumoral infiltration of cytotoxic lymphocytes [101]. To be more detailed, PCSK9 directly interacts with amyloid precursor-like protein 2 (APLP2) which literally bridges MHC-I towards lysosomal degradation [102]. All of this blocks MHC-I recycling by inducing peripheral immune tolerance. The same mechanism of promotion of lysosomal degradation by MHC-1 occurs for CD81, CD36, and the LDL receptor [103]. Consequently, the use of PCSK9i can didactically create peripheral immune tolerance against tumor cells by optimizing recognition by T lymphocytes [104]. Furthermore, the reduction of plasma levels of total cholesterol would lead to the inhibition of the ACAT-1 enzyme (deputy for cholesterol esterification) with further enhancement of the antitumor immune response [105].

These data reveal that PCSK9 may be a crucial regulator for cancer immunotherapy. Inhibitors of 3-hydroxy-3-methyl-glutaryl-coenzyme A reductase are the most important and studied pharmacological treatments aimed to decrease total cholesterol and LDL [106]. For every 38.6 mg/dl reduction in LDL-C, ASCVD events are reduced by 21% after 1 year of treatment with moderate- or high-intensity statin therapy. Recent RCTs have demonstrated that statin therapy combined with either ezetimibe or PCSK9 inhibitors reduces ASCVD events in high-risk populations [107]. As described in introduction, treatment with ICIs reduce cancer-induced immune tolerance, providing significant improvements in survival and prognosis of cancer patients at different stages of the disease [108]. PCSK9 is a new key player of cancer immune tolerance [109,110,111]. Recently, it was demonstrated that PCSK9 regulates proliferation and apoptosis in human cancer cells [112]. For example, in neuroglioma and NSCLC, knockdown of PCSK9 gene activates cancer apoptosis through caspase-3 and XIAP/p-Akt pathways [113]. Another preclinical study in melanoma-bearing mice concluded that PCSK9 gene silencing significantly increases the response to ICIs [114,115]. PCSK9 is also expressed and released by cancer cells [116,117]. It was also reported that lowering blood cholesterol levels could boost cancer immunotherapy based on adoptive T cells [118]. Cholesterol regulates the recycling of MHC I in cell membrane [119]. PCSK9 blocking agents reduce systemic levels of total cholesterol and LDL, affecting the risk of atherosclerosis but also reducing the supply of cholesterol by cancer cells [120,121] through the enhancement of apolipoprotein E receptor, CD36, β-secretase 1, and others [122,123]. A study on colon cancer [124] reveals that PCSK9 expression is upregulated in tumor cells compared with non-tumor cells and correlates with the degree of tumor invasiveness. Downregulation of the PCSK9 gene reduces colonic epithelial-to-mesenchymal transition, n-cadherin, and type 9 metalloproteases via the PI3K/AKT pathway. Furthermore, PCSK9 regulates the polarization of colonic peritumor macrophages, resulting in a pro-inflammatory and pro-metastatic phenotype [124]. PCSK9 blocking agents could therefore be studied in patients with metastatic colon cancer.

An association study of 14 lobular or ductal breast cancer patients found that systemic levels of PCSK9 are significantly higher in the advanced stage (stage III) than in patients with less aggressive disease stages and in benign lesions (*p* < 0.05). However, this study involved only a few dozen patients, so studies on a larger cohort of breast cancer patients should be conducted soon [125]. A study of colon-bearing mice treated with PD-1 blocking agents found that treatment with immunotherapy increased tissue expression and systemic levels of PCSK9, CD36, and TGF-β [126]. Combination of PCSK9 and PD-1 blocking agents in these models significantly increased the antitumor efficacy with strong synergism compared to monotherapies [126]. The use of PCSK9 blocking agents increased the levels of pro-inflammatory and immune-stimulant cytokines, of intra-tumoral CD3+CD8+ lymphocytes, reducing CD4+, FOXP3+, CD25+ lymphocyte density. In liver cancer models, researchers [127] demonstrated that PCSK9 is involved in resistance to VEGFR, PDGFR, and RAF kinases inhibitor sorafenib by direct inhibition of phosphatase and tensin homolog (PTEN) and consequent upregulation of the AKT pathway. In another study in colorectal cancel model, [128,129] authors founded that PCSK9 induces oncogenesis in APC/KRAS mutant models of colon cancer and that systemic PCSK9 levels correlate with reduced survival in this patient cohort. PCSK9 blocking agents, especially when combined with statins, inhibit the neoplastic growth of APC/KRAS mutant colon cancer xenograft models by suppressing the KRAS/MEK/ERK pathway. Thus, the investigators conclude that PCSK9 inhibition may be a valid adjuvant therapy for APC/KRAS mutant colorectal cancer by suppression of the KRAS/MEK/ERK oncogenic pathway. A recent review [130] summarizes that PCSK9 blocking agents are able to reduce prostate, lung, colon, and glioblastoma cancer cell growth through the induction of apoptosis, pyroptosis, and necrosis. Moreover, PCSK9 gene silencing reduces liver metastasis in melanoma-bearing mice through induction of intra-tumoral CD3+CD8+ lymphocytes levels [131]. In another recent meta-analysis, patients with PCSK9 loss of function mutation have a lower odds of prostate cancer compared to non-mutant PCSK9 subjects [132]. Another study on ovarian cancer models [133], evidenced that PCSK9 is also upregulated in ovarian cancer cells and correlated with tumor invasiveness by direct stimulation of ERK/MEK pathways. The use of PCSK9 blocking monoclonal antibodies or SiRNAs suppresses the growth of ovarian cancer cells by reducing AKT phosphorylation by reducing endogenous lipogenesis in these cells [133].

A clinical study of non-small cell lung cancer patients has shown that low systemic levels of PCSK9 (<95 ng/mL) predict better responsiveness to the Programmed Death-1 (PD-1) Inhibitor Nivolumab and have better overall survival than patients with higher blood levels (>120 ng/mL) [134]. In a recent biochemical study [135], an aptamer PL1 and Pcsk9 siRNA were able to potentiate anti PD-1/PD-L1 therapies in human colorectal cancer cells through enhancement of IFN-γ and Granzyme B expression. A preclinical research study concluded that PCSK9 inhibition is able to increase major histocompatibility protein class I (MHC-I) membrane density on different cancer cells through the inhibition MHC-I lysosomal degradation, thereby increasing the concentration of intratumoral cytotoxic CD3+ CD8+ lymphocytes [136].

## 5. PCSK9 Inhibitors in Cardio-Oncology: A Potential Therapy in ASCVD Cancer Patients Treated with ICIs

Immune-related adverse events (irAEs) seen in cancer patients treated with ICIs includes myocarditis, myositis, myasthenia gravis–like syndrome, endocrinophaties, and visceral obesity [137,138]. Moreover, immune-related atherosclerotic vascular events (irAVEs) are also still associated with ICIs therapy in preclinical and clinical trials. In a recently published retrospective study, ICIs therapies increased atherosclerotic plaques volume through the involvement of immune-related inflammation pathways. Other ongoing prospective studies (NCT04586894, NCT03709771, NCT04115410) are aimed to demonstrate if ICIs therapies could increase AVEs events and potential mechanisms involved [139,140,141]. However, in cardio-oncology, the reduction of several cardiovascular and cancer risk factors should be strictly considered. Among the best known cardio-oncological risk factors, we have the reduction of hyperglycemia, hypercholesterolemia, especially OX-LDL, triglycerides, homocysteine, insulin levels, hs-CRP, visceral fat, and smoking. Based on this, patients should be encouraged to follow a diet that complies with the European anti-cancer code and the WCRF directives, a non-sedentary lifestyle where possible (which is difficult especially in metastatic patients), and avoid smoking. However, PCSK9i-based drugs could be key cardioprotective strategies to reduce ASCVD in cancer patients treated with ICIs. Based on this, it is conceivable that a target population that could benefit from PCSK9 therapy are cancer patients treated with ICIs therapies, especially those with confirmed ASCVD. To this end, therefore, the benefits would be multiple: 1. enhancement of immunotherapy efficacy; 2. inhibition of mechanisms of resistance to apoptosis; 3. reduction of the risk of destabilization of stenotic plaque; 4. reduction of the risk of atherosclerosis induced by ICIs. To date, no studies have evaluated if PCSK9 blocking agents could reduce ASCVD in cancer patients treated with ICIs. A potential clinical trial could therefore include patients treated with ICIs in monotherapy or in combination with, for example, Ipilimumab, Nivolumab, Pembrolizumab, Durvalumab, Avelumab, or Atezolizumab.

## 6. Conclusions

Inhibition of PCSK9 has emerged as a novel therapy to treat hypercholesterolemia and related cardiovascular diseases. Recent preclinical and clinical evidence supports the anticancer and immune-stimulating properties of PCSK9 inhibition therapy through the inhibition of peritumoral peripheral immune tolerance and reduction of cytokines involved in cancer cell survival. Cancer patients treated with ICIs in monotherapy or in an association therapy regimen have a three times greater risk of developing atherosclerotic plaques than patients not treated with ICIs and represent a population category that requires close pharmacological monitoring of atherosclerotic risk factors, including systemic inflammation, LDL, and OX-LDL levels. Cancer patients who develop ICIs-related ASCVD could benefit from PCSK9 inhibition therapy in order to reduce atherosclerotic events, cardiovascular mortality, and improve overall survival. Therefore, cardioprotective properties of PCSK9 inhibitors should be urgently explored in randomized clinical trials in patients with cancer at high risk of ASCVD.

## Figures and Tables

**Figure 1 cancers-15-01397-f001:**
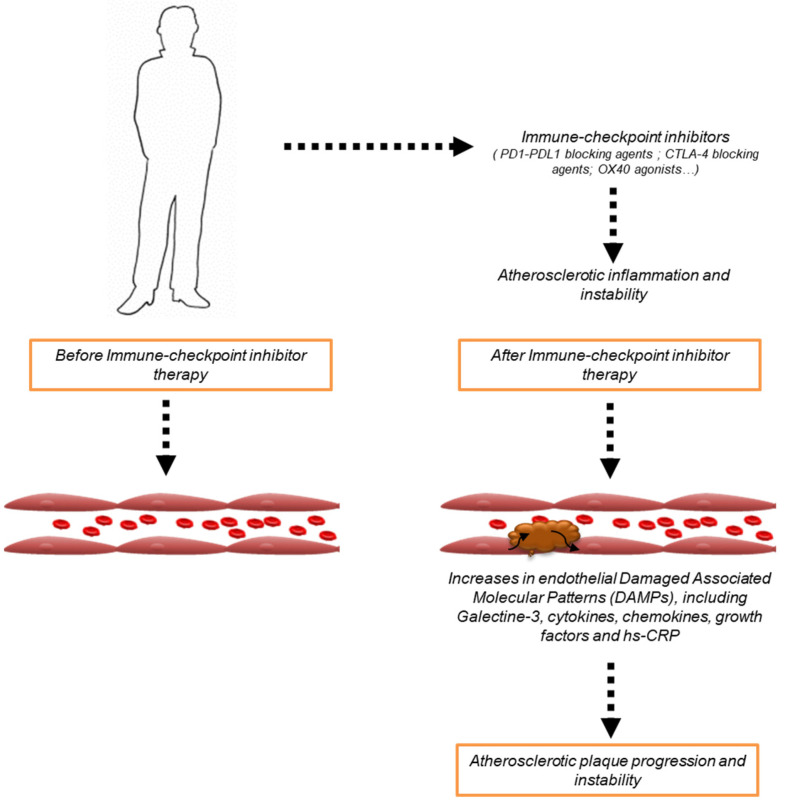
Cancer patients treated with ICIs therapies (i.e., PD-1 or PDL-1 or CTLA-4 blocking agents or OX40 agonists) increases endothelial DAMPs, including Galectine-3, cytokines, chemokines, growth factors, and hs-CRP that are associated with plaque progression and instability.

**Figure 2 cancers-15-01397-f002:**
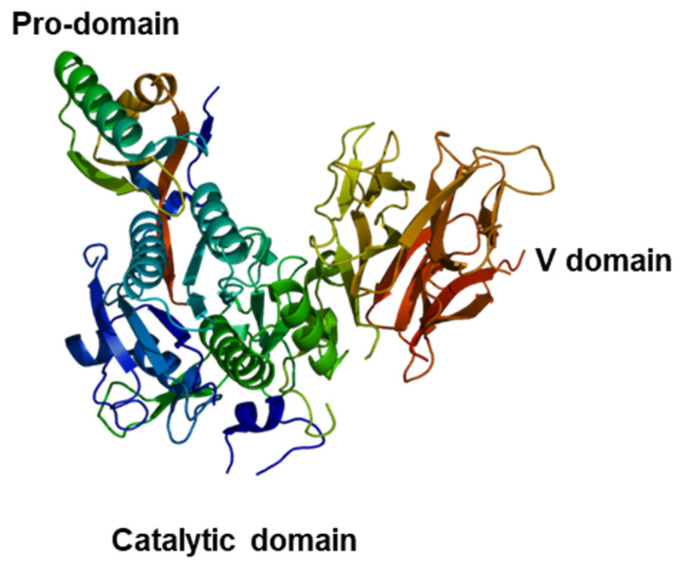
PCSK9 chemical structure. The structure of PCSK9 reveals four major components in the pre-processed protein: the signal peptide (residues 1–30); the N-terminal pro-domain (residues 31–152); the catalytic domain (residues 153–425); and the C-terminal domain (residues 426–692), which is further divided into three modules.

**Figure 3 cancers-15-01397-f003:**
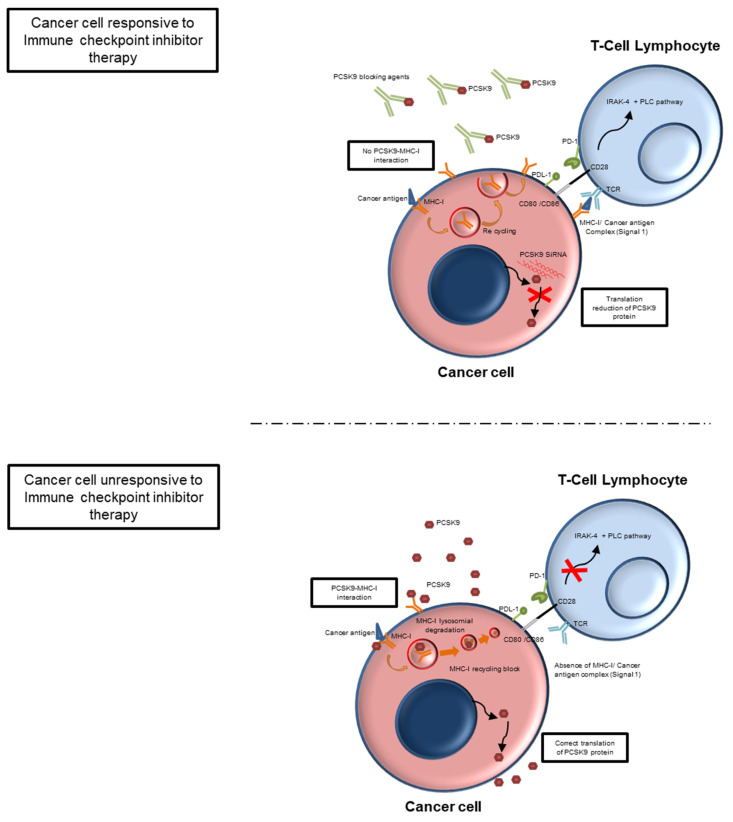
PCSK9 role in cancer immune-recognition. PCSK9 induces lysosomal degradation of MHC-I after binding to a cancer antigen, thus reducing cancer membrane MHC-I density, making the tumor cell immune-irresponsive (peripheral immune tolerance). When PCSK9 is blocked by monoclonal antibodies, MHC-I is recycled after binding to a cancer antigen, therefore allowing the recognition of the MHC-I-Antigen complex by the TCR (signal 1) which, thanks to the type 2 signal (CD28-CD8 binding), allows the activation, proliferation, and survival of the T lymphocyte and the consequent rupture of the peripheral immune tolerance.

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
