# Peer review of "PCSK9 Inhibitors in Cancer Patients Treated with Immune-Checkpoint Inhibitors to Reduce Cardiovascular Events: New Frontiers in Cardioncology"

_cancers, 2023, doi:10.3390/cancers15051397_

Round 1

Reviewer 1 Report

This is an interesting review article regarding the involvement of PCSK9 in cardiovascular risk of patients treated with ICIs and potential role of PCSK9 inhibitors in cardio-oncology. My suggestions in order to improve the manuscript.

1. All abbreviations should be explained as full text the first time they are used. Please check throughout the manuscript to correct this.

2. In the first paragraph of the Introduction, it should be underlined that cardiovasular toxicity due to ICIs is relatively rare. 

3. In the section  ICIs therapy, PCSK9 and risk of atherosclerotic cardiovascular diseases the paragraphs are too long and not well defined. Please revise the structure of the section and make sub-paragraphs.

4. This sentence: Preclinical models with deactivating mutations of PD-1 or PDL-1 or 100 CTLA-4 genes are more exposed to atherosclerotic plaques characterized by high levels of 101 VCAL-1, ICAM-1, galectine-3, oxLDL, high macrophage density and cytokines pro-in- 102 flammatory (35,36) (Figure 1).  does not correlate with Figure 1. The reader may think that Figure 1 illustrates the preclinical study where the mutations were de-activated. It should be placed there: 

These findings led to the hypothesis that ICIs could increase the risk of atherosclerotic cardiovascular diseases 106 (ASCD) in cancer patients (38) (Figure 1)

5. Section 3. PCSK9i in cardiovascular outcome trials - Please add a small table summarizing the PCSK inhibitors and their indications

6.  A potential clinical trial could include, therefore patients treated with ICIs, in mon- 391 otherapy or combination (for example Ipilimumab, Nivolumab, Pembrolizumab, Durval- 392 umab, Avelumab or Atezolizumab) 

The parenthesis is obsolete.

Author Response

This is an interesting review article regarding the involvement of PCSK9 in cardiovascular risk of patients treated with ICIs and potential role of PCSK9 inhibitors in cardio-oncology. My suggestions in order to improve the manuscript.

            -Thank you for appreciating our work and scientific quality.

  1. All abbreviations should be explained as full text the first time they are used. Please check throughout the manuscript to correct this.

            - Ok, we are agree with you and sorry for this mistake. We have carefully explained all abbreviations in text.

  1. In the first paragraph of the Introduction, it should be underlined that cardiovasular toxicity due to ICIs is relatively rare. 

            -Ok, we are in agree with you. We have updated the final version of the manuscript with a proper epidemiological analysis of ICIs-cardiotoxicity in cancer patients describing thai it is very rare

  1. In the section  ICIs therapy, PCSK9 and risk of atherosclerotic cardiovascular diseases the paragraphs are too long and not well defined. Please revise the structure of the section and make sub-paragraphs.

            - Ok, we have added therr paragraph in this section in order to make it easier for readers to understand scientifically the concepts reported. Thank you for your suggestion

  1. This sentence: Preclinical models with deactivating mutations of PD-1 or PDL-1 or 100 CTLA-4 genes are more exposed to atherosclerotic plaques characterized by high levels of 101 VCAL-1, ICAM-1, galectine-3, oxLDL, high macrophage density and cytokines pro-in- 102 flammatory (35,36) (Figure 1). does not correlate with Figure 1. The reader may think that Figure 1 illustrates the preclinical study where the mutations were de-activated. It should be placed there: These findings led to the hypothesis that ICIs could increase the risk of atherosclerotic cardiovascular diseases 106 (ASCD) in cancer patients (38) (Figure 1)

       - We are in agree with the reviewer. We have modified the manuscript in this sentance in line with your suggestion. Thank you

  1. Section 3. PCSK9i in cardiovascular outcome trials - Please add a small table summarizing the PCSK inhibitors and their indications

            - Ok, we have added a small table to make the section 3 more easy to understand. Thank you for the suggestion.

  1. A potential clinical trial could include, therefore patients treated with ICIs, in mon- 391 otherapy or combination (for example Ipilimumab, Nivolumab, Pembrolizumab, Durval- 392 umab, Avelumab or Atezolizumab) 

The parenthesis is obsolete.

  • Sorry for the mistake. Thank you for your suggestions.

Reviewer 2 Report

The publication entitled "Proprotein Convertase Subtilisin/Kexin Type 9 (PCSK9) inhibitors in cancer patients treated with immune-checkpoint inhibitors: new frontiers in cardioncology" is a very good piece of paper.

Other than minor tweaks, I don't see the point in picking on her too much.

Below are my comments.

1. The title is way too long. It needs to be redone.

2. The abstract in the first part is very misleading. It should be prescribed.

3. In the set of keywords, what do we need: PCSK9; and the other abbreviation? They are probably unnecessary and should either be removed or replaced with something.

4. Fig 1 - very poor quality - please correct it.

5. Fig 3 also - the same as fig 1.

6. Conclusion is also too weak. They should be prescribed.

7. Why almost 150 citations! Can't you do less?

Overall, it's a very nice job, but it has some flaws.

Author Response

The publication entitled "Proprotein Convertase Subtilisin/Kexin Type 9 (PCSK9) inhibitors in cancer patients treated with immune-checkpoint inhibitors: new frontiers in cardioncology" is a very good piece of paper.

Other than minor tweaks, I don't see the point in picking on her too much.

    • Thank you for your comment and useful suggestions aimed to improve the quality and readability of the manuscript. Here the point by point reply. Thank you

Below are my comments.

  1. The title is way too long. It needs to be redone.

            - Ok we are agree with you, we have changed the title to make it more direct and easy to understand.

  1. The abstract in the first part is very misleading. It should be prescribed.

                 - Ok, sorry for this mistake. We have changed the abstract and modified to make it more fluid and concentrated

  1. In the set of keywords, what do we need: PCSK9; and the other abbreviation? They are probably unnecessary and should either be removed or replaced with something.

                  - You are right. We have modified the keywords.

  1. Fig 1 - very poor quality - please correct it.

                  - Ok, we are in agree with you, we have modified the figure 1 in line with your suggestion.

  1. Fig 3 also - the same as fig 1.

                 - Ok, we are in agree with you, we have modified the figure 3 in line with your suggestion.

  1. Conclusion is also too weak. They should be prescribed.

              - Sorry for the mistake, ok we have changed the conclusion and updated in a more appropriate manner

  1. Why almost 150 citations! Can't you do less?

             - We apologize if they seem too many references, we have tried to give as much information as possible on the subject. In the current version of the manuscript we have slightly reduced the references on some redundant topics ( the last ones). Thanks again for the valuable suggestions

Overall, it's a very nice job, but it has some flaws.

    • Thank you very much for appreciating our work and for the useful tips.

Round 2

Reviewer 1 Report

The comments have been answered. The revised manuscript is now ready for publication.